# A Systematic Review of Heat Load in Australian Livestock Transported by Sea

**DOI:** 10.3390/ani8100164

**Published:** 2018-09-27

**Authors:** Teresa Collins, Jordan O. Hampton, Anne L. Barnes

**Affiliations:** College of Veterinary Medicine, Murdoch University, 90 South Street, Murdoch 6150, Australia; j.hampton@ecotonewildlife.com (J.O.H.); a.barnes@murdoch.edu.au (A.L.B.)

**Keywords:** cattle, heat stress, mortality, physiology, sheep, sea transport, stress, welfare

## Abstract

**Simple Summary:**

The transport of livestock by sea (‘live export’) is an important contemporary animal welfare issue in Australia. There is particular concern for the effects of heat load on the welfare of sheep being shipped live from Australia to the Middle East during the Northern Hemisphere summer. To reduce bias in a contentious context, we performed a systematic review of the literature relevant to Australian sea transport, heat load, and livestock. We discuss the factors contributing to harmful heat load, pathways for mitigating risks and existing knowledge gaps. We identified several areas requiring research to address these knowledge gaps.

**Abstract:**

The transport of animals by sea (‘live export’) is one of the most important current animal welfare issues in Australian society. Recent media attention has highlighted concerns regarding the effects of high environmental temperature and humidity on the welfare and mortality of sheep being shipped live from Australia to the Middle East, especially during the Northern Hemisphere summer. To improve understanding of how and why harmful heat load occurs, we systematically reviewed Australian research into heat load and sea transport. High thermal load occurs during the sea transport of sheep and cattle from Australia when animals are subject to hot and humid environmental conditions and cannot remove heat generated by metabolic processes in the body, potentially also gaining heat from the environment. Several approaches have been proposed to mitigate these risks, including avoidance of voyages in hot seasons, selection of heat-resistant livestock breeds, reducing stocking density, and improved ventilation. We identified a lack of scientific literature relating to heat load in animals transported by sea and considerable potential for bias in the literature that was found. We identified the following priority research areas: (i) experimental manipulation of variables thought to influence the incidence and severity of harmful heat load, including sheep density; (ii) further assessment of the Heat Stress Risk Assessment (HSRA) model used to predict heat load events, and (iii) development of a suite of animal welfare indicators that may allow identification of ‘at risk’ sheep before they reach debilitating heat load condition. Addressing these knowledge gaps will assist efforts to reduce the frequency and intensity of harmful heat load events.

## 1. Introduction

Few animal welfare issues have been as persistently contentious in Australia as the ‘live export’ of livestock via sea [1,2,3]. Recent media attention [4] has highlighted concerns regarding the effects of high environmental temperature and humidity on the welfare and mortality of animals being shipped live from Australia to the Middle East, especially during the Northern Hemisphere summer. These events have affected both sheep (*Ovis aries*) and cattle (*Bos taurus* and *B. indicus*), the species most commonly exported by Australia. Given the repeated occurrence of high heat load events with elevated sheep mortalities, and less common events involving cattle [5], how the industry is regulated [6,7,8] has been questioned [9,10,11]. A lack of understanding of what variables influence the likelihood of animals experiencing harmful heat load may have hindered attempts to predict and mitigate heat load events. Narrative reviews have been published on this issue in 2014 [11] and 2016 [12], and several unpublished reviews [13,14] have been produced since a media exposé in 2018 [2,4], but to our knowledge, no systematic literature reviews have previously been performed on this topic. Systematic reviews are particularly valuable in contentious situations in which bias on the part of authors may influence which studies are included and excluded (see Appendix A).

### Aims of This Review

This review attempts to synthesise Australian research on heat load in sheep and cattle exported by sea. To reduce author bias, we used a systematic search strategy and only included Australian studies. We attempted to synthesise what is known about this process and identify knowledge gaps where future studies are needed.

This is not a narrative review written to advance a political or ethical position. Narrative literature reviews are publications that describe and discuss the state of the science of a specific topic or theme from a theoretical and contextual point of view and the authors do not disclose how they chose to include studies. The authors aimed to report on the outcomes of published and unpublished animal-based studies without applying our own judgement as to what constitutes ‘good enough’ or ‘unacceptable’ outcomes for animals in the context of sea transport. This review does not attempt to assess heat load in contexts outside of sea transport and does not consider conditions outside of ships originating in Australia. This review does not attempt to assess animal welfare impacts of sea transport as a whole.

## 2. Systematic Review Methods

The authors minimised author bias by using a systematic search strategy according to the PRISMA guidelines (see Appendix A) to identify relevant journal articles, books, book sections, unpublished reports, conference proceedings, procedural documents and theses.

### Summary of Literature Reviewed

We found a total of 93 literature items matching our criteria. These comprised 51 peer-reviewed studies (55% of literature items), 29 unpublished reports, three book chapters, five theses, two conference papers, and three procedural documents. Most literature items found were contemporary, with 86% published since 2000. Further information is provided in Appendix A.

## 3. Background to Australian Sea Transport of Livestock

Australia has been the largest exporter of live agricultural animals worldwide in recent decades, with the industry valued at AUD$1659 million in 2016, comprising 304 cattle voyages and 36 ship voyages exporting over one million sheep and over one million cattle that year [15]. Sheep and cattle have been exported from Australia via sea transport for at least 30 years, predominantly to the Middle East and South East Asia. Most sheep exported live from Australia are sourced from Western Australia, and are sent to the Middle East, with the voyages taking an average of 21 days. Most cattle are sourced from Northern Australia, and are sent to South-East Asia, with the voyages taking an average of 5 days. Voyages occur year-round [15]. During these voyages, hot environmental conditions are often encountered particularly during the Northern Hemisphere summer (May–October) [16] and the adverse impact of heat on transported animals has been reported since 1989 [17,18].

### Heat Load and Sea Transport

During the sea transport of livestock from Australia to the Middle East, wet bulb temperature (WBT; an environmental measure dependent on dry bulb temperature and humidity [19]) commonly reaches 30 °C and can reach maximums between 32 °C and 34 °C with little diurnal variation [17]. Reduced diurnal fluctuation in environmental temperature during the shipment period limits an animal’s opportunity to lose heat gained during the previous day, and if heat dissipation is less than the accumulation of heat within the body, the animal accumulates a “heat load” [20,21].

Dangerous heat load may occur for transported livestock in any area experiencing high sustained environmental temperatures. This may occur in many areas of Australia, particularly Northern Australia during the Southern Hemisphere summer, and has been documented for exported animals during the Northern Hemisphere summer, at ports in the Persian Gulf [5,16,22].

## 4. Physiology of Heat Load

The basic physiology of thermoregulation has been covered in detail elsewhere [23]. Briefly, mammals such as sheep and cattle have complicated homeostatic systems to keep their body temperatures within a reasonably narrow range. The amount of heat produced by an individual of a given mammal species will be influenced by factors such as nutrition (amount, type, and timing of feeding), body size, breed, physiological status, and acclimatisation [20]. Metabolic heat and heat from the environment can increase body temperature but mammals are able to maintain their homeostatic body temperature over a wide range of ambient temperatures by balancing heat loss or gain, and heat production. There are also behavioural responses to increased temperatures, such as changing posture (e.g., stock stand or spread out to increase surface area for heat loss, reduce activity, and seek shade if outside) [24]. However, the accumulation of heat without dissipation may result in mortality or several debilitating physiological changes. Before such adverse animal welfare events occur, acute heat load episodes exhibit a predictable pattern of physiological stages [25].

### 4.1. Thermal Zones

Maunsell Australia [26] cited the thermoneutral zone (TNZ) for livestock shipping as the range of environmental temperatures at which the deep body temperature should remain constant. Within that zone, body temperature can be kept in the normal range by constant heat loss through usual sensible and insensible mechanisms. The upper limit of this zone is the upper critical temperature, and when the animal is exposed to environmental conditions above that limit, body temperature rises.

### 4.2. The Heat Stress Threshold (HST)

In the context of sea transport, the environmental conditions are considered to be best described by WBT. The upper critical WBT beyond which body temperature rises 0.5 °C above what it would otherwise have been is expressed by the live export industry as the heat stress threshold (HST) has been defined as the WBT when the core body temperature is 0.5 °C above what it otherwise would have been [26]. The same authors defined “mortality limit” (ML) as the ambient WBT above which the uncontrollable rise in deep body temperature leads to death of the animal. The environmental WBT at which body temperature rises has been the subject of observational studies [27,28], experimental research [20,29] and much debate [13]. The data sets for establishing these WBT thresholds are somewhat limited and they have been further extrapolated to cover a wider range of animals [26]. The debate which criticises the values used for HST may not adequately distinguish the complexities of the different thermal zones, the species, breed and individual differences in response to environmental conditions. The definition of HST and the use of this definition in the HSRA may not sufficiently account for the effects of environmental conditions, acclimatisation, and thermoregulatory responses of animals [12]. The concept of HST and the HSRA model also does not take into account the cumulative effects of heat load over time and the capacity of the animals to recover during periods of respite [14].

We are unaware of any literature examining the extent to which animals experience discomfort, “stress”, or distress at different body or environmental temperatures. However, some authors have attempted to relate human perceptions to how animals may be feeling e.g., [12]. Therefore, we conclude that decisions about cut-off or threshold environmental conditions have been made on physiological grounds, with less capacity to include measurements of behaviour or affective state. Clinical observations of animals subject to high environmental heat and humidity describe elevations in body temperature, with varying increases in different tissues (peripheral, rectal, core), increased heart rate, changes in peripheral perfusion, changes in respiratory rate and character, reduction in feed consumption, often an increase in water consumption, and changes in behaviour [20,29].

## 5. Assessment of Heat Load on Animals

### 5.1. Point and Cumulative Effects

The effects of heat on animals reflect both a single extreme heat insult, and prolonged cumulative effects, that is, heat load may be imposed by exposure to a short period of extreme heat, or may be the result of prolonged exposure to hot conditions, if there is no relief or cooling [21]. Australian feedlot studies have examined the effect of prolonged or chronic (110 day) heat load on cattle [30]. Duration of stress or suffering is central to considerations of animal welfare impacts generally, and this extends to livestock transport and heat load episodes in sea transport [14].

### 5.2. Recovery and Respite

Australian Merino sheep can maintain body temperature within the normal range during exposure to a prolonged increase in heat (maximum temperature of 38 °C, minimum temperature of 28 °C) and to recover quickly from the negative effect of heat load within two days of conditions returning to thermoneutral conditions [31]. Stockman, et al. [32] reported that Merino wethers experienced significant physiological changes during exposure to prolonged and continuous high heat and humidity, but maintained most aspects of homeostasis despite being hyperthermic and recovered quickly when conditions returned to thermoneutral. Stockman [20] described an additional experiment which subjected sheep to hot, humid conditions without any diurnal cooling, whereby the daily mean, maximum, and minimum core body temperatures became significantly elevated.

The experimental study of Beatty, et al. [33] used a climate control room (CCR) to simulate sea transport conditions (high temperature and humidity) using small sample sizes (six animals) of *B. taurus* and *B. indicus* cattle. The results of Beatty, et al. [33] suggested that *B. taurus* cattle experience significant physiological changes during exposure to prolonged and continuous high heat and humidity, with alterations persisting for days after the heat load conditions subsided. *B. indicus* cattle were observed to experience similar but less pronounced physiological changes. The cattle feedlot industry uses heat load index, and accumulated heat load, to predict the likelihood of adverse heat events [34]. Important in these models is the period of cooling which might provide respite from excessive heat. We were unable to find any studies that empirically assessed the duration of respite periods required to protect livestock from harmful cumulative effects of repeated episodes of heat load. This knowledge gap is likely important for understanding heat load in sea transport where there is sustained exposure to hot conditions.

## 6. Heat Load and Mortality

Livestock mortality due to excessive heat load can occur under many conditions including in feedlots and during transport. Several investigations have been performed into causes of livestock mortality in sea transport [5,35,36,37,38,39,40,41,42,43] and excessive heat load has been considered an important cause, particularly in sheep [17]. There are ongoing concerns for mortalities in sheep due to excessive heat load during sea transport [11,18].

### 6.1. Mortality Monitoring

Mortality rates on board livestock sea transport ships of 0.1–2% have typically been reported for sheep and cattle ships [15] with occasional voyages reporting much higher rates (e.g., 28.5% for cattle on one voyage) [5,22]. The proportion of these deaths that are contributed to by heat load has not always been clearly reported, with many sheep deaths appearing to be due to a combination of heat load, salmonellosis, and inanition [12]. Some authors have suggested that reported data may be unreliable due to veterinarians and livestock officers being employees of export companies and there have been allegations of under-reporting [44]. Long-term analyses have shown that livestock mortalities have been generally decreasing since data began being reported in 1995. Recent analyses reveal that sheep deaths are increased during the Northern Hemisphere summer. Evidence from shipments from 2005 to 2014 is that mortality rates rise to approximately double when sheep are transported from Australia in winter to the Middle East in summer. For example, monthly mortality rates for shipments of wethers from Fremantle to the Middle East increase from approximately 0.5% (February) to approximately 1% (August) [15]. A five-year average of total mortality rate of sheep shipped from Australia to the Middle East shows mortality rates for sheep exported to the region are higher when sheep are loaded in May to October. While this pattern may only be an association, there is an “enduring stability of seasonal difference” of mortality rate in all classes of sheep over time [15]. Additional factors influence the seasonal effect on mortality rates such as in any given year, there is variability between ports with respect to average annual mortality rate [16].

### 6.2. Monitoring Outcomes other than Mortality

Mortality is one of many adverse animal welfare events whose frequency may be quantified in monitoring programs and represent the extreme end of suffering that may occur due to heat load [45,46]. For this reason, monitoring of adverse events beyond mortality has been suggested by many authors and McCarthy [14] recently recommended that the industry moves away from using mortality as a measure to a focus on measures that reflect the welfare of the animal including those that reflect heat load. Additional careful monitoring of animal behaviour at the pen-level, such as panting, eating, and resting behaviour of stock should be pursued. These data should be combined with basic environmental measures, also at the pen-level, such as temperature, relative humidity, and measures of ventilation [47].

## 7. Factors Affecting Heat Load In Transported Livestock

Considerable research has been devoted to investigating the influence of several controllable variables on heat load experienced by transported animals, including diet [48,49], ventilation [50], and environmental conditions [51].

### 7.1. Environment-Based Factors

Environmental conditions with sustained high temperatures predispose animals to harmful heat load. When hot conditions also include low diurnal temperature fluctuations, animals are further predisposed to harmful heat load. This limits an animal’s opportunity to lose heat gained during the previous day and can compound the effects of subsequent heat load. Humid conditions further reduce the capacity for animals to lose gained heat. Hot, humid conditions with low diurnal fluctuation are often encountered during the voyages of livestock transport ships to the Middle East in the Northern Hemisphere summer [13].

### 7.2. Management-Based Factors

Altering the physical environment may reduce the heat load on animals. For instance, provision of shade will reduce heat gain from solar radiant heat gain; in shipping this is only of relevance where there are open decks and animals may be exposed to direct sunlight. Radiant heat gain from hot metal infrastructure may maintain temperatures so there is little respite for animals housed within; there are currently limited options for cooling the ships other than wetting with cooler water [52].

#### 7.2.1. Stocking Density

Stocking density or space allowance is an important factor underlying heat load, and of prominent concern for livestock welfare. Incoming air from mechanical ventilation accumulates additional heat and moisture from the animals, generated through their metabolic processes. The HSRA uses in its calculations a “wet bulb rise” which describes mathematically the contribution of animal metabolic heat to the environmental conditions experienced by the animals in the ship [53]. Therefore, the stocking density of animals (head per square meter) has a strong influence on the heat load experienced by animals. Aspects such as the underlying metabolic rate and body size of animals will affect how much heat they release and metabolic rate is also affected by how much they eat. The capacity of the ventilation system to remove waste gases, such as carbon dioxide and ammonia, may further influence stocking density for each deck and area [53,54]. The expiration of carbon dioxide is dependent on the animal’s metabolic rate and the substrate metabolised, such that higher stocking density of animals may generate more carbon dioxide than can be adequately removed from an area. Ammonia in shipping is primarily generated through chemical breakdown of urea in urine and faeces (and bedding) and therefore may reach greater concentrations with higher animal density and increased waste production [55].

#### 7.2.2. Ventilation

The role of ventilation in sea transport has been investigated by industry-funded research projects [50,53,56]. Livestock transport ships may either have open decks (natural ventilation) or closed decks with mechanical ventilation [53]. Most livestock vessels rely on mechanical ventilation, which serves three main purposes. Firstly, it replenishes air (including oxygen). Second, it removes heat. Third, it removes waste gases, including water vapour (evaporates moisture from the manure pad), carbon dioxide, and ammonia [14]. Ammonia is a highly irritating alkaline gas that has been associated with adverse effects on sheep on transport vessels [57,58]. Air movement is considered to be important and airspeed can be used to give an ‘adjusted WBT’ [53]. On ships equipped with forced ventilation systems, pen air turnover (PAT) and speed of air flow are two aspects of air movement which are considered within management models regarding carriage of livestock [27,53,56]. The mechanical ventilation systems currently used on livestock vessels work on high air turnovers (e.g., 50 m/h or m^3^/h divided by m^2^) which are required to remove gases and assist in removing moisture from faecal pads [53]. Increased flow of cooler, drier air will enhance convective and evaporative heat loss. If the air is hotter than the animals, or saturated with moisture, the cooling effect is diminished, and hot, humid air may contribute to heat gain rather than heat loss.

It has been proposed that a risk management approach may be required for operations involving open deck pens with no mechanical ventilation [53]. Ships require mechanical ventilation of open decks only when the breadth is greater than 20 m, otherwise they rely on natural ventilation. McCarthy [14] has recommended that all vessels should be re-certified to determine pen air turnover, air speed, and ventilation patterns, before travelling to the Middle East during the Northern Hemisphere summer.

#### 7.2.3. Provision of Feed and Water

Feed intake can be influenced by heat load [59] and feed type can influence heat production. Nutritional management of animals in hot conditions can involve reducing total energy input, through provision of feeds with higher roughage content or by restricting total feed intake [60]. It is common management practice to introduce feed restrictions in high-risk heat load conditions in order to limit excessive heat output from animals due to digestion [61]. Time of feeding can also altered to coincide with the cooler part of the day. Kennedy [61] performed a CCR experiment that investigated the effect of different grain feeding approaches on cattle (16 steers) subjected to heat load over three days. He reported that when environmental heat load was imposed in a CCR, cattle fed a wheat diet showed greater thermal stress than cattle fed a sorghum diet, but when animals were subjected to a second period of heat load, the result was equivocal [61].

Provision of supplements may assist animals in responding to the heat [62,63]. It has been suggested that employing a dietary supplement may be a cost-effective and simple method for ameliorating the negative impact of heat load in sheep [48]. Electrolyte supplementation of cattle under hot conditions is proposed to assist with the acid-base changes that occur due to panting. Beatty, et al. [49] provided electrolytes in feed and water to 80 *B. taurus* steers on a livestock transport ship and reported higher live weights in the supplemented cattle. While some degree of heat load was observed during the trial, the steers were not considered clinically heat stressed during the experiment. It was not apparent from CCR research performed with sheep that electrolyte supplementation was similarly beneficial [20], although there may be acid-base and electrolyte changes in extreme conditions.

Supplementation with supra-nutritional doses (beyond required levels) of Vitamin E and selenium may ameliorate effects of high heat load in sheep [64,65,66,67,68]. Betaine (trimethylglycine), an amino acid capable of acting as an organic osmolyte or a methyl donor, can improve animal production measures in cattle, pigs, poultry and lambs and has been suggested as a useful supplement for heat load management in sheep [48]. Drinking water temperature may affect heat load in livestock. Offering chilled water to sheep [69] may be a useful method to decrease body temperature during times of high heat load, although it has been shown that sheep and cattle will drink greater volumes of warm water [69]. Further research is required to determine if electrolyte supplementation for cattle and nutritional supplementation for sheep would be beneficial and feasible for on-ship use when animals endure periods of high head load.

#### 7.2.4. Management of Pens Including Bedding and Manure Pad

Provision of bedding is linked to ventilation and air quality [70] and may influence heat load experienced by transported livestock. In Banney, et al. [71] the link between air quality and bedding is described, whereby ventilation will affect the moisture content of the bedding, and the removal of noxious gases produced in the bedding. It is important that the bedding does not contribute to further production of heat or noxious gases, such as ammonia or carbon dioxide, as might occur when organic matter ferments. A variety of materials have been tested or used for bedding in animal industries, including sawdust, straw, woodchips, pine shavings, and desiccated manure [70]. Sawdust is the most frequently used material for cattle and is required on ‘long-haul’ (>10 day) voyages.

The manure pad from sheep is generally quite dry, and if it remains firm, dry and intact, it is considered by the industry to be the preferred choice of bedding material for sheep during sea transport [1,55,71]. However, if the sheep manure pad becomes excessively wet, it can contribute to problems with the production of noxious gases [55], with sheep having difficulty moving around (“pugging”), and with faecal contamination of the legs and body of the sheep. When there is high environmental heat, with increased humidity and increased urine output from sheep drinking more, the ventilation may not be able to keep the manure pad sufficiently dry, exacerbating the problems [71]. The manure from cattle being more liquid than that from sheep means generally the cattle pens need more regular cleaning during long haul voyages, although this might not be necessary during short haul voyages. Banney, et al. [71] describe in detail the processes around washing down the cattle pens, and the cattle themselves, with the addition of new sawdust after the washing. They note the advantages of washing and wetting the cattle in providing some cooling relief during very hot conditions, but underline the essential role of good ventilation at that time in limiting a rise in humidity.

### 7.3. Animal-Factors

#### 7.3.1. Breed Selection

Animal factors can be manipulated to ameliorate the adverse effects of heat load on livestock. Specifically, selection of breeds arising from hot regions over those evolved in temperate regions generally improves tolerance to heat load. For instance, *B. indicus* cattle (originating from South Asia) generally have greater heat tolerance than *B. taurus* cattle (originating from Europe) [46,72,73,74,75,76]. Omani, Niamey or Awassi sheep breeds (Middle Eastern) similarly have greater heat tolerance than Australian Merino sheep (originally from Europe) [20,77]. While *B. indicus* cattle are abundant in Northern Australia [46], the availability of Middle Eastern sheep breeds in Southern Australia, when compared to Merino flocks, is much lower. As such, changing sheep breeds in sea transport could only be a long-term strategy that would require economic modelling as it would impact on farm profitability.

#### 7.3.2. Acclimatisation

Acclimatisation of animals to heat requires exposure to hot conditions for several days or weeks [76,78]. With respect to live export, this means acclimatisation to heat or cold should be in place before the transport process commences ideally, so that the animals are prepared in a climate similar to which they are travelling. During that time, there will be behavioural and physiological responses that decrease metabolic heat production, such as decreased feed intake and metabolic rate, and other responses that improve their ability to lose heat, such as increased sweating, and higher plasma volume [29].

#### 7.3.3. Effect of Fleece on Sheep

Experimentally, shearing has been shown to significantly increase the heat tolerance of rams, presumably by enhancing the efficiency of evaporative cooling from the skin [79]. Anecdotal reports have suggested that recently shorn sheep cope better than fleeced sheep with hot conditions encountered during the sea transport voyages to the Middle East. As a consequence, sheep destined for sea transport may be shorn in the immediate period before shipping, to limit wool cover and so improve heat loss [80]. Beatty, et al. [81] tested this hypothesis with a CCR experiment involving shorn and fleeced Merino sheep. They found that fleeced sheep maintained higher core and rumen temperatures and respiratory rates than shorn sheep under all environmental conditions. Maunsell Australia [50] reported that when WBT was >26 °C on livestock transport ships, unshorn ewes were hotter than shorn ewes by 0.2 °C to 0.4 °C as measured by rectal temperature.

There are concerns that pre-embarkation shearing may contribute to increased stress, and inappetence. To address these concerns, an experiment was performed whereby 600 sheep were fitted with Radio Frequency Identification tags, and subsets were shorn each day (days 1, 2, 3, 4 or 5) and time and frequency of feed and water trough attendance were determined [80,82]. There was no difference in time spent at feed or water troughs between any treatment groups on any day, and minimal behavioural changes were observed. This suggests that shearing may occur on any day during the pre-embarkation feedlot period, and that current management practices regarding shearing do not disrupt time spent feeding.

## 8. Pathways for Reducing Excessive Heat Load in Livestock Sea Transport

Risk assessment approaches have been developed and refined by the sea transport industry for anticipating conditions likely to precipitate heat load episodes [5,51,83]. The response variable traditionally underlying this approach has been animal mortality but it has recently been proposed to replace the mortality limit with a heat tolerance level within the risk assessment model [14]. Risk assessment approaches have also been developed for heat load management in feedlot cattle [84]. Critics of the Australian government’s risk assessment approach have argued that the estimate of the heat stress threshold of sheep used in the model is substantially higher than that observed under simulated sea transport conditions, which may lead to an underestimate of the importance of heat load in sheep on voyages where mortality is high [12]. It is widely recognised that further improvements are required to reduce the incidence of harmful heat load episodes for exported sheep [14]. Suggested pathways for reducing the incidence and severity of harmful heat load episodes for exported livestock are listed below from most drastic to most subtle.

### 8.1. Avoidance of Seasons and Extreme Weather Events

Proposals have been made that sea voyages should be avoided in the Northern Hemisphere summer. Scientific reviews have suggested that there is elevated risk to sheep exported from Australia during summer in the Middle East and ethical arguments have been advanced that this risk is sufficient to warrant consideration of restriction of trade during this period [12,13].

#### Heat Load Forecasting

Considerable research has been devoted to forecasting heat load for animals in feedlots [84,85,86]. Heat load forecasting within the Australian feedlot industry has evolved over a period of two decades [87]. These developments have been used by the sea transport industry to generate a computer model that aims to assess the risk of heat stress and to contain mortality levels on livestock ships below certain arbitrary limits. The Heat Stress Risk Assessment (HSRA) model ‘HotStuff’ model was developed for the Australian livestock export industry to estimate and minimise the incidence of heat stress mortality in livestock during voyages to the Middle East [26]. The model has been in operation since 2003 and has had several refinements and reviews [51,83]. The HSRA model provides a framework from which to address heat stress and heat load. The model factors in the weather (both predicted and actual for the destination ports), the type, class and body weight of animals, and ship factors such as ventilation design and airflow. The latter requires input from the vessels’ design specifications and is adjusted for each vessel prior to loading [26].

However, despite the sophistication of the model, it often fails to accurately predict a heat load event [87]. The accuracy of the HRSA model is currently being questioned, and the ability to prevent animals experiencing harmful heat load when travelling to the Middle East may be limited by the ventilation capacity of the vessels. Given mounting recent evidence of ongoing heat load events affecting exported sheep, it is argued that a review of the settings used for the input of data in addition to a review of the risk settings should be undertaken [14]. The current settings used is a 2% probability of a 5% mortality due to heat stress, and was chosen by industry [51,83]. It has recently been suggested that the risk setting should be replaced by the likelihood of an animal experiencing heat stress, not mortality, in order to achieve improved welfare outcomes [14].

It has also been suggested that revision of the HSRA model should include consideration of the different heat load thresholds, with examination of the appropriateness of using 0.5°C rise in body temperature, the WBT at which that occurs, and the duration of that episode, as an arbitrary threshold [13]. Phillips [12] criticised the HST values used within the HSRA model as being above those suggested by animal studies. It is important to note that some of the discussion in recent reviews regarding heat stress thresholds (HSTs) do not appear to fully appreciate the distinctions between how different researchers have expressed the thresholds. For example, Stockman [20] described three different HSTs. The HSRA further iterates ML for different classes of stock, from a base animal of specified type, weight, and normal body temperature. Ferguson, et al. [51] did not criticise the methodology or base values in their review of the model.

The HSRA model does not yet have the capacity to deal with the effects of cumulative heat load. It would be advisable for future revisions of the model to use expertise from the feedlot industry to consider the influence of duration of heat exposure and the capacity for respite to influence effects of heat load.

### 8.2. Improved Ventilation

The issue of ventilation is central to heat load events in sea transport. There are several pathways that have been proposed for refinement of current conditions. Ships with ‘open decks’ (lacking mechanical ventilation) may pose extra challenges for improving air flow during high-risk heat load conditions [50,53,56]. Simple maintenance steps have been proposed to ensure that existing ventilation systems are operating at maximum capacity. First, removal of anything that obstructs airflow in and around the decks can be performed. Second, regular checking of fans to ensure they are working at full capacity can be performed. Third, it is important to ensure that all exhaust outlets are free of obstruction [14].

#### Air Conditioning

The approach of installing air conditioning in livestock transport ships has been considered as an avenue for cooling animals during high-risk conditions. However, air conditioning requires a low air turnover (and often recycling of air) in order to be effective (and/or cost-effective). Low air turnover systems would likely be less effective at removing waste gases and faecal moisture from animal pens, and for this reason, with current available technology, air conditioning does not seem to be a viable option for livestock vessels [14].

### 8.3. Reducing Animal Density

It has been argued that currently used stocking densities on sheep voyages are too high [13,14], contributing to too great a WBT rise, especially when the prevailing environmental conditions are already close to the HST. The HSRA model uses stocking density as a critical factor in determining WBT rises across the deck. As detailed above, there is little scientific literature to allow accurate elucidation of threshold densities that would safeguard cattle and sheep across all voyages, and the HSRA has extrapolated data for risk management. However, given that stocking density is the key parameter in managing livestock in long haul voyages to the Middle East, sizable reductions in allowable stocking densities have recently been suggested [13,14]. Further studies should be undertaken to examine the effects of such changes across stock classes. The use of allometric equations [88,89] to determine space allocation of livestock undergoing sea transport has been proposed, and further work is required to determine the appropriate *k* values for use in allometric equations to appropriately limit the wet bulb rise. Reducing stocking density may also improve other animal welfare issues such as the ability of animals to cope with ship motion [90].

### 8.4. Improving Bedding

Exposure to high environmental heat causes animals to drink and urinate more [32,33], which then increases moisture in the pens. Excess moisture in the sheep faecal pad can be managed by providing additional sawdust, a useful interim measure to ensure that sheep don’t get bogged down or have fleece covered with faecal material [70,71]. Exact ventilation rates to ensure adequately dry faecal pads have not been determined given these will vary with climate, pen location, ship type and sheep density. The careful management of hosing in cattle pens has also been suggested to improve bedding during times of high heat load [52].

### 8.5. Wetting

Wetting of animals can be used to mitigate high heat loads via lowering the temperature of the microclimate surrounding animals. This approach has been used for cattle in feedlots and on livestock ships [52]. Wetting has been trialed and has been shown to lower peak body temperatures and hasten recovery from hyperthermia. Four water application methods (hosing, overhead sprinklers, leg sprinklers, and misters) were evaluated by Tait [52] in a CCR experiment with *B. taurus* cattle, and high-pressure hosing was found to be the most effective. It is unknown how practical water spray cooling would be on-board ships or how effective it would be for sheep. There are several concerns about the use of wetting for sheep. Wetting has the effect of increasing local humidity, causing undesirable wetting of the faecal pad and wetting the fleece of sheep which will only slowly dry in humid conditions. For these reasons, wetting has not been further explored for preventing harmful heat load in transported sheep but is considered to be a useful approach for cattle [52].

## 9. Conclusions

Commercial transport of livestock by sea from Australia has been conducted for over 30 years, and although there has been continual improvement in mortality incidence [15,91], high mortality events associated with heat load continue to occur. Moreover, public concern for the welfare of transported animals is elevated since a 2018 media exposé on the negative effects of heat load [2]. Despite the imperative for objective understanding of this condition, we found some potential for author bias in the literature. There has been insufficient independent science (55% of literature items we found were peer-reviewed) which addressed heat load and sea transport of livestock, with much of the literature consisting of non-peer-reviewed reports funded by industry and narrative reviews written to express opposition to the industry. Animal-based studies indicate that harmful heat load is often observed in livestock transported by sea from Australia, particularly in sheep and cattle sent to the Middle East in the Northern Hemisphere summer. For the industry to be socially sustainable, further scientific investigation is required to identify avenues for reducing the incidence of harmful heat load events. The extent to which avoiding consignments in the hot summer months may be sufficient to avoid the majority of incidents of heat stress remains untested.

Trade in live animals can bring substantial financial benefits for exporters and importers [92], but it requires the trust and goodwill on the part of both trading partners as well as community support in democratic countries. Improved understanding of the effects of heat load on livestock, when extreme heat load occurs, and ways to prevent its occurrence are required to minimise animal welfare impacts for livestock transported by sea. Eliminating the risk of extreme heat load would likely require reductions in stocking densities, improvements to ship ventilation and on-board stock management, and possibly ceasing shipments to the Middle East in the Northern Hemisphere summer. These factors combined may mean that some transport consignments become uneconomic, highlighting the difficulty of balancing animal welfare and economics in this context. In states like Western Australia, where the sheep industry is underpinned by live export, such trade interruptions could have deleterious impacts on industry viability.

Prudent suggestions from what is known include moving away from using mortality as the main determinant of animal welfare outcomes [14] and development of multiple animal-based parameters. Ongoing adverse heat load events suggest that further review of the currently used HRSA is vital. Studies that determine the duration of respite periods required to protect animals from harmful cumulative effects of heat load are essential. In addition, studies that can describe and validate a list of welfare indicators that reflect the physical and affective state of animals should be considered [93]. In the first instance, careful monitoring of animal behaviour at the pen-level, such as panting, eating and resting behaviour of stock should be pursued, combined with basic environmental measures, such as temperature, relative humidity, and measures of ventilation [47].

Concurrently, studies in land-based facilities that can examine direct effects of changing one variable at a time (e.g., varying stocking density in CCR experiments) will be informative. Given this type of research will take time for relationships between factors to be described, interim measures could be commenced immediately to reduce the short-term incidence of harmful heat load from existing knowledge. Such measures may include reducing stocking density, for instance using allometric principles, providing adequate bedding to ensure a consistently firm faecal pad, and providing more detailed monitoring of livestock. There are likely to be important interactions between ship factors (ventilation), management (stocking rate, fodder, bedding), and animal (body size, weight, breed) factors. The importance of each of these will vary according to voyage length and climate. Unless pathways can be found to reduce the incidence and severity of heat load episodes, and to demonstrate these improvements in refinement of animal welfare outcomes, the community support and sustainability of this form of livestock transport may well expire in Australia.

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
