# Peer review of "A Systematic Review of Heat Load in Australian Livestock Transported by Sea"

_animals, 2018, doi:10.3390/ani8100164_

Reviewer 1 Report

The authors are to be congratulated for tackling a  very difficult subject. There is however opportunity for improvement by improving the clarity.   Some sections would require an existing knowledge of the industry by the reader, for them to understand the context and meaning as written.  More detailed comments are included in the attached document. 

Author Response

Thank you for your review. Attached  is a description of how we have addressed your comments.

Reviewer 2 Report

The review entitled “A Systematic Review of Heat Load In Australian Livestock Transported by Sea” synthesize Australian research on heat load in sheep and cattle exported by sea. The transport of livestock by sea representing an important topic for animal welfare issue in Australia. In particular it's important to evaluate the effects of heat load on the welfare of sheep transported during hot season. This is an important review in order to minimize stressful condition during the transport by sea.

The reference should be improved.

So, the study is very interesting but I think that the manuscript is acceptable for publication after minor changes.

The manuscript would benefit from rewriting of large section and review by a native English speaker.

Specific comment

Title: reflects the topic’s review

Key words: delete “transport” and rewrite “Sea transport”; delete “stress” and rewrite “social stress”

1. Introduction:

Expand this part by indicating some data on the number of livestock transported to Australia by sea.

Please inserted some reference on transport in livestock:

Navarro G., et al.,.(article in press) Effects of space allowance and simulated sea transport motion on behavioural and physiological responses of sheep. Applied Animal Behaviour Science.

 Fazio F., et al., 2018. Livestock handling and road transport influence some oxidative stress parameters in ewes. Journal of Veterinary Behavior, 2018, 26, 5-10.

1.1. Aims of This Review; 1.2. Definitive scope of This Review

I do not understand the reason to write two different aim in this review. it is better to insert a single aim that includes everything so as not to confuse readers

2. Systematic Review Methods; 2.1. Summary of Literature Reviewed

Pag 4 : please delete lines 64-72, it is not important to describe the quantity of references that has been consulted for this review. The references is shown in the specific section.

7.3.3. Fleece in Sheep

Include some research on the physiology of thermoregulation in sheep and on the importance of fleece in the thermoregulatory response

Casella S., Giudice E., Passantino A., Zumbo A., Di Pietro S., Piccione G. (2016). Shearing induces secondary biomarkers responses of thermal stress in sheep. ANIMAL SCIENCE PAPERS AND REPORTS, 34, p. 73-80.

Piccione G., Casella S., Alberghina D., Zumbo A., Pennisi P. (2010). Impact of shearing on body weight and serum total proteins in ewes. SPANISH JOURNAL OF AGRICULTURAL RESEARCH, vol. 8, p. 342-346, ISSN: 1695-971X

4. Physiology of Heat Load

Please add this references:

Tresoldi G., et al. 2018. Cooling cows with sprinklers: Spray duration affects physiological responses to heat load. Journal of Dairy Science. 101, 4412-4423.

I suggest to insert a new paragraph: “Biomarkers of transport stress” it’s important to prevent transport stress through a new hematological and hematochemical biomarkers:

Inserted:

Fazio et al., 2015. Utility of acute phase proteins as biomarkers of transport stress in ewes and beef cattle. Italian Journal of Food Safety, 2015, 4: 4210.

Mahabub A., et al.,2018.  Assessment of transport stress on cattle travelling a long distance (≈648 km), from Jessore (Indian border) to Chittagong, Bangladesh. Veterinary Record Open 5: e000248. doi: 10.1136/vetreco-2017-000248

Please add Conclusion with A Way Forward

Author Response

Thank you for your review. We have detailed in the attached how we have addressed your comments,
